# *In vitro* assembly of plasmid DNA for direct cloning in *Lactiplantibacillus plantarum* WCSF1

**Marc Blanch-Asensio©, Sourik Dey©, Shrikrishnan Sankaran[iD]\***

Bioprogrammable Materials, INM—Leibniz Institute for New Materials Campus D2 2, Saarbrücken, Germany

© These authors contributed equally to this work.
\* Shrikrishnan.sankaran@leibniz-inm.de

**Data Availability Statement:** All raw data including images and sequencing files related to results described in this paper have been added to the OSF data repository and can be accessed at this DOI - DOI: 10.17605/OSF.IO/C6X3D.

## Abstract

Lactobacilli are gram-positive bacteria that are growing in importance for the healthcare industry and genetically engineering them as living therapeutics is highly sought after. However, progress in this field is hindered since most strains are difficult to genetically manipulate, partly due to their complex and thick cell walls limiting our capability to transform them with exogenous DNA. To overcome this, large amounts of DNA (>1 μg) are normally required to successfully transform these bacteria. An intermediate host, like *E. coli*, is often used to amplify recombinant DNA to such amounts although this approach poses unwanted drawbacks such as an increase in plasmid size, different methylation patterns and the limitation of introducing only genes compatible with the intermediate host. In this work, we have developed a direct cloning method based on in-vitro assembly and PCR amplification to yield recombinant DNA in significant quantities for successful transformation in *L. plantarum* WCFS1. The advantage of this method is demonstrated in terms of shorter experimental duration and the possibility to introduce a gene incompatible with *E. coli* into *L. plantarum* WCFS1.

## Introduction

Lactobacilli are a group of Gram-positive bacteria of great importance to the food and healthcare industries with numerous strains identified as being beneficial for humans, and used as probiotics [1–3]. Furthermore, since they naturally colonize almost every site of the human body that hosts a healthy microbiome, *e.g.* the gastrointestinal tract [4,5], urogenital tracts [6], oral cavity [7] and nasal cavity [8], lactobacilli are an excellent foundational candidate for the development of live biotherapeutic products (LBPs) [9]. Beyond their natural health benefits, there is considerable interest in engineering them with heterologous genes for therapeutic applications like drug delivery [10,11] and mucosal vaccinations [12,13]. However, one of the crucial factors slowing down progress in lactobacilli engineering is difficulties in transforming them with exogenous DNA [14]. This is largely due to their thick and complex cell wall structures, which prevent successful bacterial transformation if the concentration of plasmid DNA is less than >1 μg [15]. To obtain such high plasmid DNA quantities, shuttle vectors are often used that can be amplified in intermediate hosts, predominantly *E. coli* [16]. To facilitate the construction of recombinant plasmids, several shuttle vectors have been identified, which can

**Funding:** This work was supported by a the Deutsche Forschungsgemeinschaft (DFG) Research grant [Project # 455063657 - https://gepris.dfg.de/gepris/projekt/455063657] for M.B. A., the DFG Collaborative Research Centre, SFB 1027 [Project # 200049484 - https://gepris.dfg.de/gepris/projekt/466932240] for S.S. and the Leibniz-Gemeinschaft's Leibniz Science Campus on Living Therapeutic Materials [LifeMat - https://www.lsclifemat.de/] for S.D. The funders had no role in study design, data collection and analysis, decision to publish, or preparation of the manuscript.

**Competing interests:** The authors have declared that no competing interests exist.

undergo stable replication in both the cloning host, *E. coli* and the relevant Lactobacilli strains [17–19]. Nevertheless, since *E. coli* is a Gram-negative bacterium that is phylogenetically distant from Lactobacillus genera, this strategy can lead to genetic sequence incompatibilities due to GC-content differences [20], DNA methylation [21], repetitive sequence insertions [22] and toxic protein buildup in the *E. coli* cloning host [23]. Alternatively, the Gram-positive lactic acid bacterium, *Lactococcus lactis*, can also be used as an intermediate host for recombinant plasmid construction [24]. However, the availability of functional replication origins in *L. lactis* is limited [25] and inclusion of additional broad-range replicons significantly increases the size of the plasmid. The excessive increase in the size of the plasmid might lead to segregational instability [26] and thereby limit the size of the heterologous genes that can be included in it. Hence, it is desirable to be able to directly transform circular plasmid dsDNA into the lactobacilli strains without relying on intermediate bacterial hosts like *E. coli* and *L. lactis*. To avoid the need for an intermediate host, Spath et al. developed a direct cloning approach based on the assembly of PCR-amplified DNA fragments by restriction digestion and ligation to obtain optimal quantities of circular dsDNA for transformation in *Lactiplantibacillus plantarum* CD033 [27]. They further show that the unmethylated plasmid DNA allowed for transformation in a strain (*L. plantarum* DSM20174) that could not be transformed using methylated DNA, possibly due to native restriction-modification systems. However, the method still requires the presence of restriction sites within the DNA sequences, which can limit the versatility of combining heterologous genes in the plasmid and needs to be accounted for when dealing with strains that may harbor unknown restriction-modification systems.

In this work, we report a direct cloning method that leverages the Gibson assembly strategy and takes advantage of recent advances in cost-effective oligonucleotide synthesis. By doing so, we avoid the need for restriction sites and improve the feasibility of combining diverse DNA sequences to construct versatile recombinant plasmids. We demonstrate this direct cloning method in *Lactiplantibacillus plantarum* WCFS1, one of the commonly engineered probiotic *Lactobacillus* strains for which improved engineering methods are highly sought [28,29]. Furthermore, this direct cloning method is considerably quicker in comparison to indirect cloning methods requiring an intermediate host. We have characterized the efficiency and accuracy of this approach and have demonstrated the successful cloning of a gene expressing the medically relevant protein, Elafin which showed a high failure rate when being cloned through the intermediate host, *E. coli*. Thus, this direct cloning method will be instrumental in enabling the cloning of Lactobacilli with a wider variety of heterologous genes and with greater versatility than previously possible.

## Materials and methods

### Bacterial strains and growth conditions

*L. plantarum* WCFS1 was used as the parent strain in this study. The strain was grown in the De Man, Rogosa and Sharpe (MRS) media (Carl Roth GmbH, Germany, Art. No. X924.1). Recombinant *L. plantarum* WCFS1 strains were grown in MRS media supplemented with 10 μg/mL of erythromycin (Carl Roth GmbH, Art. No. 4166.2) at 37˚C and 250 rpm shaking for 16 h. For the indirect cloning experiments, NEB 5-alpha Competent *E. coli* cells were used (New England Biolabs GmbH, Germany, Art. No. C2987). This strain was grown in Luria-Bertani (LB) medium (Carl Roth GmbH, Art. No. X968.1). Recombinant *E. coli* DH5α strains were grown in LB media supplemented with 200 μg/mL of erythromycin at 37˚C, 250 rpm shaking conditions for 16 h.

## Molecular biology

Q5 High Fidelity 2X Master Mix (New England Biolabs GmbH [NEB], Germany, No. M0492S) was used to perform DNA amplification. Amplified DNA products were purified using the Wizard® SV Gel and PCR Clean-Up System (Promega GmbH, Germany, Art. No. A9282). Foragarose gels, 1 kb Plus DNA Ladder(Catalog Number 10787018) and Generuler 100 bp Plus DNA Ladder (Catalog Number (SM0321) was procured from ThermoFisher Scientific™, Germany and used for reference. Primers were synthesized by Integrated DNA Technologies (IDT) (Louvain, Belgium) and the elafin gene fragment was ordered as eBlock from IDT (Coralville, USA). All primers used in this work are shown in S1 Table in S2 File. The mCherry gene fragment was amplified by PCR from a plasmid previously generated in our lab. The genetic sequences of mCherry and elafin genes are shown in S2 Table in S2 File. The plasmid *pLp3050sNuc* (Addgene plasmid # 122030) [30] was used as the vector backbone in this study. The schematic for the recombinant plasmids, pLp_mCherry and pLp_elafin constructed in this study have been highlighted in S1 Fig in S2 File. The Codon Optimization tool from IDT (Choice Host Organism–*L. acidophilus*) was used to optimize the codon bias for mCherry coding segment. The Java Codon Adaptation Tool (JCat) [31] was used to codon-optimize the gene encoding for the human peptidase inhibitor 3, elafin (GenBank ID: AAX36874.1) using the codon optimization database for *L. plantarum* WCFS1. DNA assembly was performed using the HiFi Assembly Master Mix (NEB GmbH, Germany, Art. No. E5520S). For plasmid circularization, the Quick Blunting Kit (NEB GmbH, Germany, Art. No. E1201S) and the T4 DNA Ligase enzyme (NEB GmbH, Germany, Art. No. M0318S) were used.

## *L. plantarum* WCFS1 electrocompetent cell preparation

Wild-type *L. plantarum* WCFS1 was cultured overnight in 5 mL of MRS media and at 37°C with shaking (250 rpm). After 16h, 1 mL of the culture ($OD_{600}$ = 2) was added to 20 mL of MRS media and 5 mL of 1% (w/v) glycine. This secondary culture was incubated at 37°C and 250 rpm until the $OD_{600}$ reached 0.8. The cells were then harvested by centrifugation at 4000 rpm (3363 X g) for 12 min at 4°C. After discarding the supernatant, the pellet was washed twice with 5 mL of ice-cold 10 mM $MgCl_2$. The pellet was then washed twice with ice-cold Suc/Gly solution (1 M sucrose and 10% (v/v) glycerol mixed in a 1:1 (v/v) ratio), first with 5 mL and second with 1 mL. Next, the supernatant was discarded, and the bacterial pellet was resuspended in 450 µL of ice-cold Suc/Gly solution. Finally, 60 uL aliquots were prepared and immediately used for DNA electroporation or stored at -80°C for future use.

## Electroporation based transformation in *L. plantarum* WCFS1

For electroporation transformation, plasmids were first mixed with 60 µl of electrocompetent cells at quantities (300–1200 ng) specified in the Results section. After a short incubation on ice, the mixture was transferred to ice-cold electroporation cuvettes with a 2 mm gap (Bio-Rad Laboratories GmbH, Germany, #1652086). Electroporation was performed using the Micro-Pulser Electroporator (Bio-Rad Laboratories GmbH, Germany), with a single pulse (5 ms) at 1.8 kV. Immediately after the pulse, 1 mL of MRS media was added to the bacterial mixture and then transferred to a 1.5 mL Eppendorf tube for further incubation at 37°C, 250 rpm for 3 h. Following this incubation, cells were pelleted down at 4000 rpm (3363 X g) for 5 min. 800 µL of the supernatant was discarded, and the remaining 200 µL was used for cell resuspension by slow pipetting. Finally, the resuspended pellet was plated on MRS Agar plates supplemented with 10 µg/mL of Erythromycin, and plates were incubated at 37°C for 48 h for colonies to grow.

## Direct cloning method in *L. plantarum* WCFS1

This study created and optimized a novel direct cloning method based on amplifying and circularizing in vitro-assembled gene fragments to be directly transformed in *L. plantarum* WCFS1 (Fig 1). For the Gibson HiFi Assembly reaction, complementary overhangs were included by PCR using a set of primers that contained the corresponding overhangs at the 5' ends. In the Gibson HiFi assembly reaction mixture, 50 ng of the PCR-amplified linear vector with overlapping DNA fragments and 10 ng of the corresponding eBlock were mixed along with 10 μl of the HiFi DNA Assembly Master Mix (Mili-Q water was added up to 20 μl). The reaction was incubated at 50°C for 30 minutes. After that, 5 μL of the assembled product was used as template for an additional round of PCR, using a set of primers that annealed specifically to the insert region. The final volume of this PCR was 120 μl, and the amplification cycle threshold was set at 22. 5 μl of this reaction was run on an agarose gel to confirm amplification (S2 Fig in S2 File). After purifying the linear PCR product, 3500 ng of DNA were phosphorylated using the Quick Blunting Kit. This reaction was performed as suggested in the standard reaction protocol. 2.5 μl of the 10X Quick Blunting buffer and 1 μl of the Enzyme Mix were mixed with the purified DNA (3500 ng). Milli-Q water was added up to 25 μl. The reaction was incubated for 30 minutes at 25°C to allow the blunting reaction and then kept at 70°C for

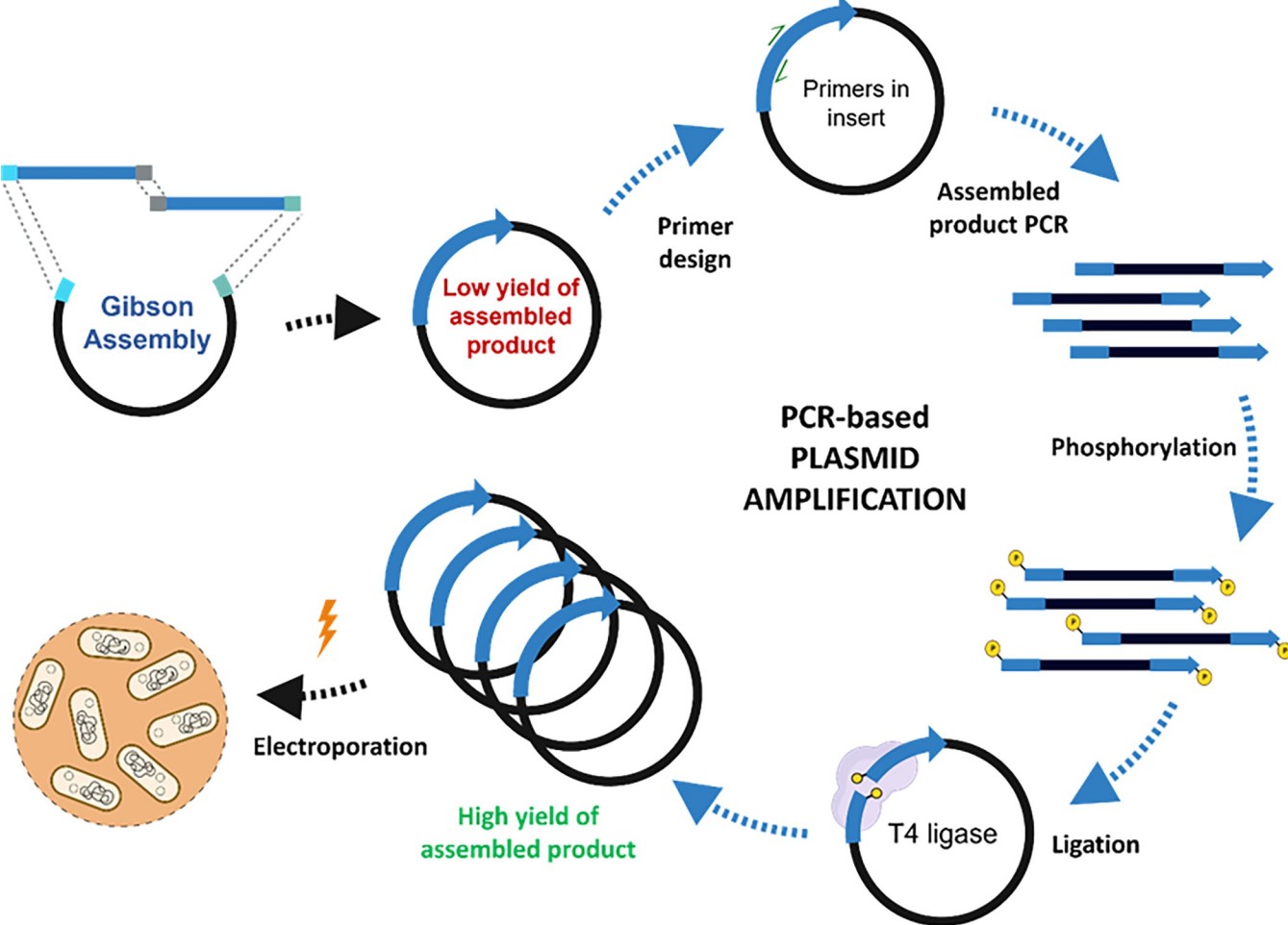

**Fig 1. Scheme of the PCR-based plasmid amplification in the direct cloning method.** The scheme was generated using BioRender.

10 minutes to inactivate the enzyme. In the next steps, phosphorylated DNA was ligated using the T4 ligase enzyme. This reaction was slightly modified from the standard protocol because a higher amount of DNA was added to the reaction. Per ligation, 500 ng of phosphorylated DNA (3.6 µl of the Quick Blunting reaction) were mixed with 1.5 µl of T4 Ligase enzyme and 2.5 µl of 10X T4 Ligase Buffer. Milli-Q water was added up to 25 µl. The number of ligation reactions depended on the amount of DNA intended to be circularized. The ligation reactions were incubated for 2.5 hours at 25˚C and then 10 minutes at 70˚C to inactivate the enzyme. Following the incubation, the ligation mixtures were combined, and the ligated dsDNA was purified using the Promega kit. In this purification, DNA was eluted 3 times with 9 µl of Milli-Q water each time to obtain the highest DNA concentration. The DNA concentration of the ligated mixture was measured (absorbance at 260 nm) using a NanoDrop Microvolume UV-Vis Spectrophotometer (ThermoFisher Scientific GmbH, Germany). The purified ligated products were then transformed into *L. plantarum* WCFS1 electrocompetent cells.

The protocol described in this peer-reviewed article is published on protocols.io, https://dx.doi.org/10.17504/protocols.io.ewov1o82olr2/v1 and is included for printing as S1 File with this article.

Sequence verification was performed by PCR amplification of the target gene conducted directly from the bacterial pellet. To do so, the selected bacteria were inoculated in MRS media supplemented with 10 µg/mL of erythromycin and incubated overnight at 37˚C and 250 rpm. The following day, 1 mL of the bacterial culture was collected in a 1.5 mL Eppendorf tube and centrifuged at 4˚C for 3 min at 8400 X g. The supernatant was discarded, the residual pellet fraction was scratched off with a sterile pipette tip and used as a template for the PCR (100 µl as final reaction volume, 28 cycles). Alternatively, a colony grown on the MRS plate can also be used to for PCR amplification of the target gene segment. The PCR settings involved an additional initial denaturation step for 10 minutes at 98˚C to ensure maximum bacterial lysis. After the PCR, 5 µl of the PCR products were assessed by agarose gel electrophoresis to confirm amplification at the expected size. Next, the PCR product was purified, and the DNA concentration was measured using the Nanodrop. Finally, 2000 ng of the purified PCR product was sent for sequencing to Eurofins Genomics GmbH (Ebersberg, Germany). An additional DNA purification step before Sanger sequencing was employed for obtaining efficient results (Additional Service: PCR Purification).

## Indirect cloning via the intermediate host strain *E. coli*

For the indirect cloning, an additional Gibson HiFi reaction was performed (identical to the Gibson HiFi reaction set for the direct protocol). However, an additional step was done before setting this reaction, which involved the restriction enzyme digestion with DpnI (NEB GmbH, Germany, Art. No. R0176S). In this reaction, 500 ng of the purified PCR product were mixed with 1 µl of DpnI enzyme and 1 µl of rCutSmart buffer (Milli-Q water was added up to 10 µl). Incubation was performed for 30 minutes at 37˚C followed by 10 minutes at 70˚C. The digested product was used for the Gibson HiFi Assembly reaction. Once the reaction was done, it was transformed into NEB 5-alpha Competent cells (50 µl). In this transformation, NEB 5-alpha Competent cells were first thawed on ice for 10 minutes. After that, 8 µl of the reaction were properly mixed with the competent cells by pipetting and incubated on ice for 20 minutes Following the incubation, a 60-second heat shock was performed by placing the cells at a 42˚C water bath. Next, cells were again incubated on ice for 5 minutes. After that, 950 µl of SOC media was added to the cell mixture and kept for incubation for 1 hour at 37˚C. Finally, 150 µl of the culture was plated on an LB agar plate supplemented with 200 µg/mL of erythromycin and incubated at 37˚C overnight.

For the pLp_mCherry cloning, the screening of positive clones was done using the Gel Documentation System Fluorchem Q (Alpha Innotech Biozym Gmbh, Germany). Bacterial colonies expressing mCherry were imaged in the Cy3 channel (Exλ/Emλ = 554 nm/568 nm) and the corresponding brightfield image was taken using the ethidium bromide channel (Exλ/Emλ = 300 nm/600 nm). One red colony was inoculated in LB media supplemented with 200 µg/mL of erythromycin and incubated at 37˚C overnight. The following day, plasmid extraction was performed using the Plasmid extraction miniprep kit (Qiagen GmbH Germany, Art. No. 27104). The plasmid DNA concentration was measured using the 260 nm absorbance setting on the NanoDrop Microvolume UV-Vis Spectrophotometer. The recombinant plasmid was then transformed into *L. plantarum* WCFS1 electrocompetent cells.

For the pLp_elafin cloning, 20 colonies were streaked on a fresh LB agar plate supplemented with 200 µg/mL of erythromycin and incubated at 37˚C overnight. The following day, positive clones were screened by PCR using a forward primer that annealed to the vector and a reverse primer that annealed to the elafin gene (100 µl as final reaction volume, 28 cycles). In the PCR, 10 minutes at 98˚C were set for the initial denaturation of the samples. Three positive clones confirmed by colony PCR were inoculated in LB media supplemented with 200 µg/mL of erythromycin and incubated at 37˚C overnight. The following day, the respective plasmids were extracted and sent for sequencing to Eurofins Genomics GmbH (Ebersberg, Germany).

## Results and discussion

The Gibson assembly approach to combining DNA fragments requires the fragments to contain terminal overhangs that complementarily overlap by ~20 bases. The method was originally developed to stitch together the first artificial genome due to the high level of flexibility it provided compared to restriction digestion-based methods [32]. It is a single reaction assembly method that can be performed without thermal cycling and within an hour. In this work, we employ Gibson reaction to conduct in-vitro assembly of circular dsDNA constructs for direct cloning in *L. plantarum* WCFS1. As shown in Fig 1, our method involves PCR amplification of a vector and an insert with overlapping arms, followed by their Gibson reaction-based assembly that yields a low quantity (50–80 ng) of the assembled dsDNA. To obtain optimal concentration of dsDNA for transformation in Lactobacilli, a second amplification and recircularization step was performed, yielding >1 µg of the desired construct. This method was characterized and optimized in terms of transformation efficiency, accuracy, and capability for cloning challenging genes in *Lactiplantibacillus plantarum* WCFS1. Different plasmid constructs were assembled and compared with indirect cloning using *E. coli* as an intermediate host.

### Transformation efficiency and accuracy of the direct cloning method

Transformation efficiency (TE) indicates the extent to which cells can take up DNA from the extracellular space and express the genes encoded by it [33]. While it is possible to transform *L. plantarum* WCFS1 with extracellular DNA, TE is typically poor [27]. To demonstrate this, a simple plasmid construct, pLp_mCherry, with gene sequences ideal for indirect cloning through *E. coli* was used. This plasmid consisted of a p256 replicon, an erythromycin resistance cassette and the mCherry gene driven by a strong constitutive promoter P$_{tlpA}$, [34] all of which are compatible in both *E. coli* and *L. plantarum* WCFS1. The plasmid was constructed through Gibson reaction-based assembly of the pLp-3050sNuc plasmid backbone and P$_{tlpA}$-mCherry insert and transformed in *E. coli*. One correctly sequenced plasmid, extracted from an *E. coli* DH5α clone, was transformed in *L. plantarum* WCFS1 at different concentrations (300, 600, 900 and 1200 ng), yielding low TE values that increased with higher DNA concentrations (20–

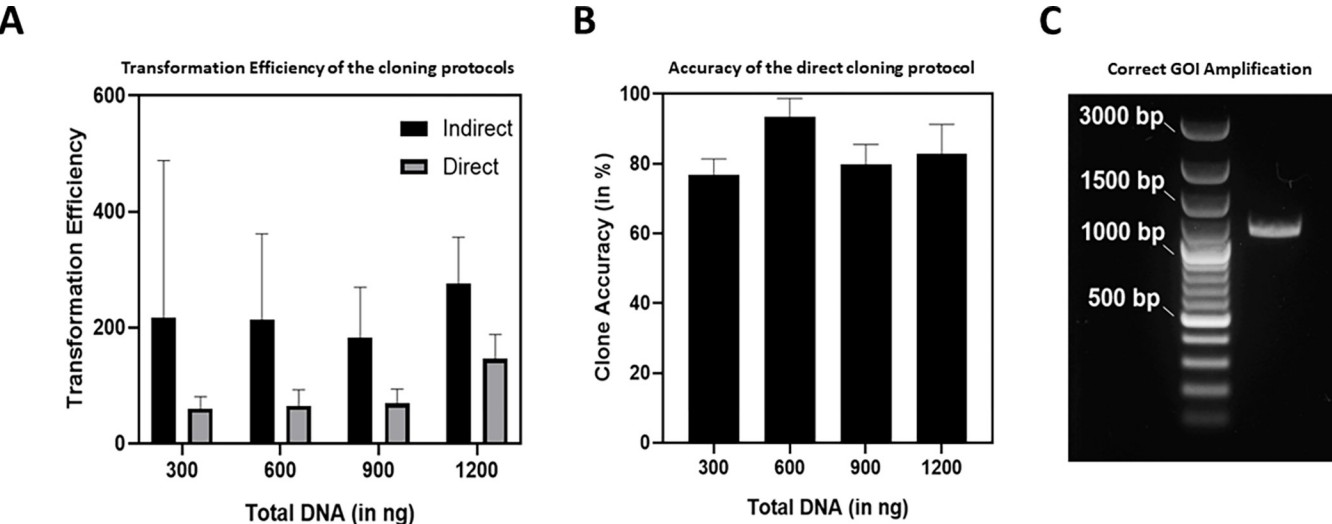

**Fig 2.** **(A)** Transformation Efficiency comparison against the total concentration of DNA transformed for the indirect and direct cloning protocol in *L. plantarum* WCFS1 (The standard deviations correspond to three independent biological replicates) **(B)** Percentage accuracy of correct recombinant clones against the total concentration of DNA transformed in *L. plantarum* WCFS1 using the direct cloning protocol (The whiskers correspond to values from three independent biological replicates). **(C)** Agarose gel showing the colony PCR product (1140 bp) corresponding to the mCherry gene of interest (GOI). A red *L. plantarum* colony obtained after direct cloning was used as the template DNA for the PCR reaction. Generuler 100 bp Plus DNA Ladder (ThermoFisher Scientific^TM) was used for the reference.

500 cfu/μg) (Fig 2A). In the direct cloning method, the Gibson reaction based assembled dsDNA was PCR amplified using complementary primers within the insert region and the resulting linear fragments were circularized by phosphorylation and ligation to yield high amounts of plasmid DNA suitable for transformation in *L. plantarum* WCFS1 (> 3μg). Following the transformation of the circularized plasmid mix in *L. plantarum* WCFS1 at different DNA concentrations as mentioned above, TE values were found to be lower than that of the indirect method. However, despite being lower the overall TE values were within the same order of magnitude and increased drastically when the net DNA concentration used for transformation was above 1 μg (Fig 2A). The lower TE values could be due to incomplete circularization of the PCR-amplified plasmid fragments, due to which the final quantity of the circularized constructs might have been lower than the total DNA that was quantified [35]. The accuracy of clones generated from the direct cloning method, determined by their ability to express mCherry, was estimated after checking the fluorescence of 418 colonies that grew across all the DNA quantities tested. Colonies were streaked on fresh plates and the following day they were examined for the presence of fluorescent protein. Overall, 347 out of 418 colonies were red, giving an accuracy of 83% (Fig 2B). The accuracy in the indirect cloning method with the correctly sequenced recombinant plasmid isolated from *E. coli* DH5α was found to be above 99%, as expected.

Furthermore, we wanted to prove that a DpnI digestion prior to the Gibson HiFi Assembly reaction would not have an impact on the number of positive clones obtained through the direct cloning method since we assumed that the backbone vectors used as template for the PCR would be extremely diluted during the multiple purification steps involved in the protocol. Therefore, we repeated the pLp_mCherry direct cloning with and without a DpnI digestion of the insert and vector PCR products prior to the Gibson HiFi Assembly reaction. We screened 10 red colonies for both experimental conditions by PCR amplification of a partial gene segment within the mCherry reporter. The proportion of red colonies obtained was

similar in both conditions and colony PCR amplification of a part of the mCherry gene (10 red colonies from each) yielded PCR products of the same expected size from all clones. This confirmed that a DpnI digestion prior to the HiFi Assembly reaction is not necessary (S3 Fig in S2 File).

Notably, since *L. plantarum* WCFS1 contains 3 endogenous plasmids [36], sequencing-based verification of desired regions in the recombinant plasmid was done by PCR amplification of the entire mCherry gene of 1140 base pairs (bp) (Fig 2C). The gene segments were directly amplified from bacterial cell pellets, and the amplicon sequencing was outsourced to an external service provider, Eurofins Genomics GmbH, where their additional DNA purification option was employed. An initial purification of the PCR amplified product by us seemed to improve the quality of the sequencing chromatograms but was not absolutely necessary to get the correct results (S5A and S5B Fig in S2 File). As expected, all clones expressing mCherry yielded the correct sequences without any mutations or deletions. The whole mCherry gene was also amplified from 10 non-red colonies using the same set of primers and PCR conditions as in Fig 2C. We obtained amplification for 7 out of 10, nevertheless, the PCR product was either bigger or smaller than expected (S4 Fig in S2 File). These results suggest that mutations might have occurred during the PCR amplification steps or a minor proportion of wrongly assembled products were formed during Gibson assembly which can result in recombinant clones with the mutated gene of interest (GOI).

## Time requirement for the direct and indirect cloning methods

The direct cloning method is considerably quicker and less labor-intensive than the indirect cloning method. All steps in the direct cloning method can be completed in 4 days after which PCR-amplified sequences can be sent for sequencing. In contrast, the indirect cloning method requires 5 to 6 days, depending on the time allocated for growth of bacteria on the master plate (Fig 3). Note that a master plate is needed to be made from transformed *E. coli* colonies in our case due to the use of erythromycin as the antibiotic resistance marker. *E. coli* has natural resistance to this antibiotic at low dosages which can be surpassed by supplementing the growth media with higher concentration of erythromycin. On one hand, this will prevent the growth of non-transformed cells but on the other hand it will also lead the transformed colonies to grow slowly making the colonies too small for reliable use in colony PCR analysis. In the case of *L. plantarum* WCFS1, a master plate was not required since colonies grown for 48 hours were large enough to handle both colony PCR analysis and inoculation in liquid cultures.

## Direct cloning of a gene incompatible with *E. coli*

The main advantage of the direct cloning method is demonstrated in the ability to transform genes in lactobacilli that are challenging using the indirect method. Genes encoding proteins that are toxic to *E. coli*, for example, often result in mutations or complete deletions of the GOI in the plasmid when transformed into *E. coli* [37]. We therefore tested the cloning of a plasmid containing the human peptidase inhibitor 3 gene, known as elafin, encoded downstream of a strong constitutive promoter ($P_{tlpA}$). This protease has been reported to exert anti-microbial activity with *E. coli*, so its constitutive expression is expected to be toxic [38,39]. Transformation of the assembled plasmid containing the constitutively expressed elafin in *E. coli* yielded very few colonies. The screening of positive clones were done by PCR amplification using a primer set, where one specifically annealed to the vector and the other one was complementary to the insert region. Only 3 clones showed amplification, but the amplified product was shorter than expected (524 bp) (Fig 4A). Sequencing of plasmids extracted from these clones revealed several mutations and deletions. The whole $P_{tlpA}$ was deleted from all three plasmids, and two

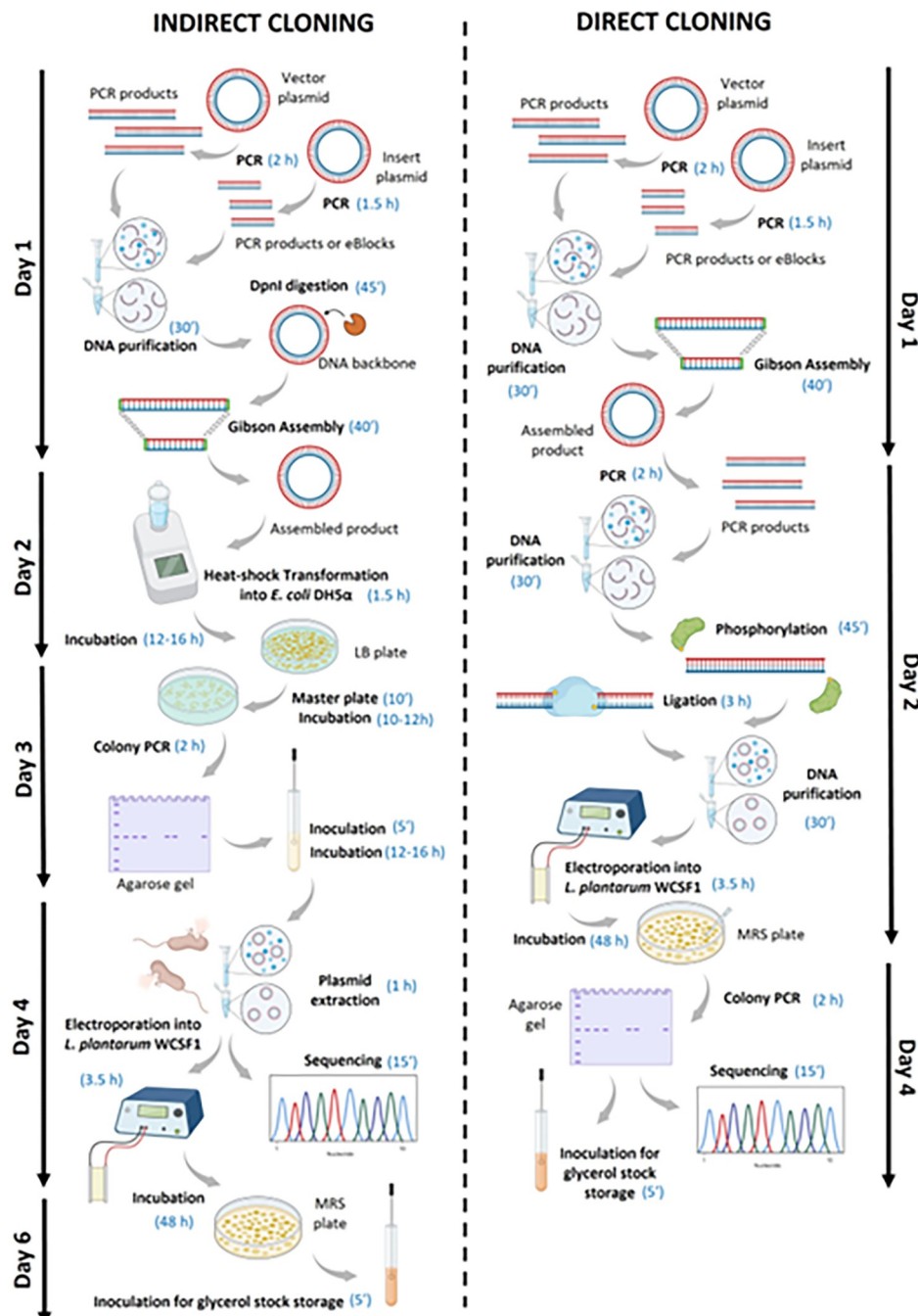

**Fig 3. Schematic representation of the steps and temporal requirements for the direct and indirect cloning methods used for *L. plantarum* WCFS1.** The scheme was generated using BioRender.

clones had the elafin coding sequence truncated (Fig 4B, S6 Fig in S2 File). On the other hand, the direct cloning method yielded over 124 colonies after transformation with 1000 ng of phosphorylated and ligated dsDNA. 10 colonies were screened by PCR, and all of them showed amplification at the expected size of 524 bp (Fig 4A). Sequencing of the gene amplified from randomly selected 3 clones revealed no mutations or deletions (Fig 4B).

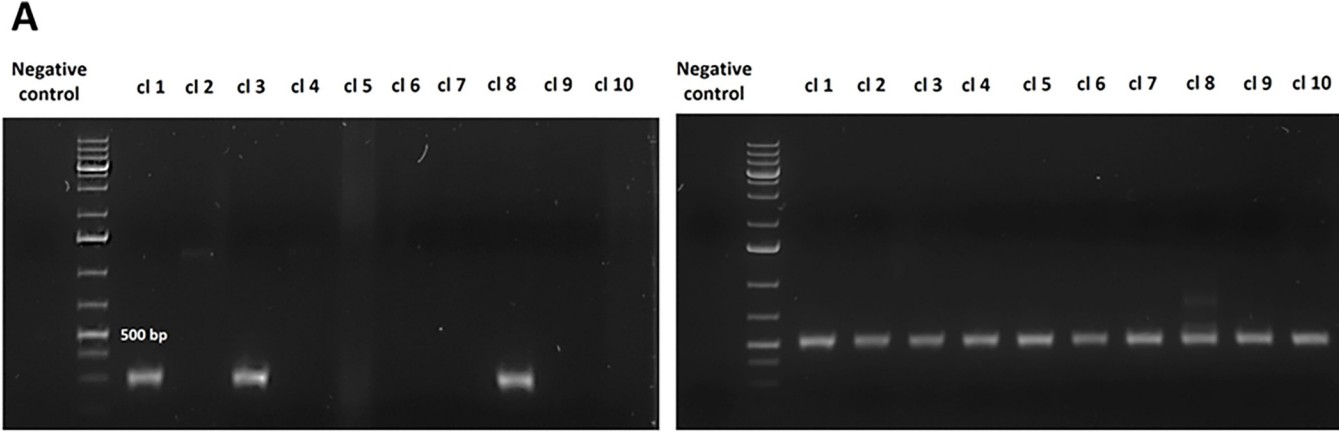

**Fig 4. (A)** The agarose gel (left) shows the colony PCR result of 10 randomly selected E. coli pLp_elafin clones. The agarose gel (right) corresponds to the colony PCR of 10 randomly selected L. plantarum WCFS1 pLp_elafin clones. All the colony PCR reactions were performed under the same conditions. The expected amplicon size for the pLp_elafin clones were 524 bp. The reference ladder used was the 1 kb Plus DNA ladder (ThermoFisher™) **(B)** Table listing the results obtained after Sanger Sequencing of the isolated pLp_elafin plasmids. The (✓) suggests a complete match to the expected sequence. Three plasmids per cloning method (direct and indirect) were sent for analysis.

## Conclusions

The direct cloning method developed in this paper has proved effective in transforming circular dsDNA (plasmid) DNA into *L. plantarum* WCFS1 without any intermediate host requirement for plasmid amplification. We demonstrate that this method provides two major benefits for lactobacillus engineering–(i) it saves time of at least 2 days compared to commonly used indirect cloning methods involving intermediate hosts and (ii) enables the cloning of genetic constructs that might be toxic or incompatible with the intermediate host. Since this method relies on PCR-amplification based *in vitro* assembly of DNA fragments, it must be noted that the accuracy can be affected by mutations that occur during PCR amplification and the possible formation of unspecific assembly fragments. To minimize the risk of mutations, a high-fidelity polymerase (Q5 DNA polymerase) was used in this study [40]. To accelerate the identification of positive colonies, a visible reporter like mCherry can be included or colony PCR can be performed. Using these methods, we confirmed that the accuracy of the transformed clones was above 80%. While we have tested this direct cloning method only in *L. plantarum* WCFS1, we believe this strategy can also be expanded to other hard-to-transform lactobacilli,

in which similar plasmids have been previously transformed using the indirect method [41]. When testing the direct cloning method on different strains, it is important to note that success will depend on whether they accept unmethylated DNA. Furthermore, if transformation is hindered by restriction-modification systems in these strains, DNA design strategies can be employed to overcome this challenge [42]. Finally, while we have used modest-sized plasmid (<4 kb) with a low copy number replicon (P256 replicon, copy number 3–5). Based on previous studies [43], it is expected that bigger plasmids with higher copy number replicons can be transformed using the direct cloning method although further investigations are definitely needed to test the effect of plasmid size on transformation efficiency and accuracy of the transformed clones.

## Supporting information

**S1 File. Supporting information file containing the step-by-step protocol generated in this study.**
(PDF)

**S2 File. Supporting information file containing all supporting information tables and figures.**
(DOCX)

## Acknowledgments

The plasmid *pLp3050sNuc* was a kind gift from Prof. Geir Mathiesen.

## Author Contributions

**Conceptualization:** Marc Blanch-Asensio, Sourik Dey, Shrikrishnan Sankaran.

**Data curation:** Marc Blanch-Asensio, Sourik Dey.

**Formal analysis:** Marc Blanch-Asensio, Sourik Dey.

**Funding acquisition:** Shrikrishnan Sankaran.

**Investigation:** Sourik Dey.

**Methodology:** Marc Blanch-Asensio, Sourik Dey.

**Project administration:** Shrikrishnan Sankaran.

**Resources:** Shrikrishnan Sankaran.

**Supervision:** Shrikrishnan Sankaran.

**Visualization:** Marc Blanch-Asensio, Sourik Dey, Shrikrishnan Sankaran.

**Writing – original draft:** Marc Blanch-Asensio, Sourik Dey, Shrikrishnan Sankaran.

**Writing – review & editing:** Shrikrishnan Sankaran.

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
