## [Decision Letter · Decision Letter 0]

6 Dec 2022

PONE-D-22-25198Gibson Assembly-based direct cloning of plasmid DNA in Lactiplantibacillus plantarum WCSF1PLOS ONE

Dear Dr. Sankaran,

Thank you for submitting your manuscript to PLOS ONE. After careful consideration, we feel that it has merit but does not fully meet PLOS ONE’s publication criteria as it currently stands. Therefore, we invite you to submit a revised version of the manuscript that addresses the points raised during the review process. Manuscript has been reviewed by 2 subject experts and both have appreciated the work that can be published. However, both have raised some important concerns and have suggested action from authors. Please go through their comments carefully and submit the revision after completely addressing the reviewer's concerns.

We look forward to receiving your revised manuscript.

Kind regards,

Hari S. Misra

Academic Editor

PLOS ONE

Journal Requirements:

“This work was supported by a Research Grant from the Deutsche Forschungsgemeinschaft (DFG)  [Project # 455063657 - https://gepris.dfg.de/gepris/projekt/455063657] for M.B.A., the Collaborative Research Centre, SFB 1027 [Project # 200049484 - https://gepris.dfg.de/gepris/projekt/466932240]  for S.S. and the Leibniz Science Campus on Living Therapeutic Materials [LifeMat - https://www.lsclifemat.de/]  for S.D.”

-----

Reviewers' comments:

Reviewer's Responses to Questions

**Comments to the Author**

1. Does the manuscript report a protocol which is of utility to the research community and adds value to the published literature?

Reviewer #1: Yes

Reviewer #2: No

2. Has the protocol been described in sufficient detail?

To answer this question, please click the link to protocols.io in the Materials and Methods section of the manuscript (if a link has been provided) or consult the step-by-step protocol in the Supporting Information files.

The step-by-step protocol should contain sufficient detail for another researcher to be able to reproduce all experiments and analyses.

Reviewer #1: Yes

Reviewer #2: Partly

3. Does the protocol describe a validated method?

Reviewer #1: Yes

Reviewer #2: Yes

4. If the manuscript contains new data, have the authors made this data fully available?

Reviewer #1: Yes

Reviewer #2: Yes

**5. Is the article presented in an intelligible fashion and written in standard English?**

Reviewer #1: Yes

Reviewer #2: Yes

6. Review Comments to the Author

Reviewer #1: The manuscript by Asensio et al, is really a fantastic example on the optimization of cloning methodologies to improve both cloning and transformation efficiencies with a wide range of heterologous expression and biochemical applications. The manuscript is very clear and well written. I recommend the manuscript for publication after some minor comments have been addressed.

The introduction is well written but some more introductory information on recently developed similar cloning strategies and how this method might provide further advantages (for example, increased DNA yield, ability to skip E. coli for toxic genes etc).

The authors do not do a DpnI digestion during the direct cloning approach and it is possible for the original template DNA to make it to the beginning of their transformation step (Day 2). Although the majority of this will be lost during the subsequent purification and PCR steps, did the authors have any problems with the background empty template being transformed? Did they find empty vectors in their colony screen? Could the authors comment on the proportion of positive vs negative clones during their colony screening and subsequent verification of plasmids in the final stages of their experiments?

The schematic figure (Figure 3) is great to show an overview of both methods. Could the authors also include slightly more information such as the strain they are performing their transformation into (cloning strain/expression strain etc). This should help to simplify the advantages of one method over another.

The authors are essentially performing PCRs on PCR products and thus the higher number of cycles eventually leads to the increased chance of incorporating mutations. We use Q5 a lot and understand that it is high fidelity and that mutations are very infrequent, also that the authors performed sequencing to ensure no errors. However, I think it would be nice if an extra sentence or two was placed in the text to ensure it is clear to both the authors and future scientists performing this method are aware of the issue and how it should be handled.

I think the conclusion is a bit weak and the authors could really use this opportunity to highlight the advantages of the method over what is currently available – particularly to the great biotechnology/synthetic biology fields.

Reviewer #2: An article by Blanch-Asensio et al. “Gibson Assembly-based direct cloning of plasmid DNA in Lactiplantibacillus plantarum WCSF1” describes a modified method to deliver plasmids to L. Plantarum. This manuscript needs major revisions before it can be published.

1. The title of the paper gives off the impression that the authors successfully demonstrated in vivo DNA assembly. But the paper is just about obtaining higher plasmid copy numbers before the transformation.

2. Introduction is lacking some information/explanations.

a. Some sentences are not clear “Hence, it is desirable to be able to clone these lactobacilli without the need for an intermediate host” - deliver plasmids?

b. Why rolling circle amplification is not described? The is a great example of using this method for creating synthetic minimal cells - https://www.science.org/doi/10.1126/science.aad6253. How would this method compare with the method described in this manuscript?

3. Results and discussion

a. “As expected, all clones expressing mCherry yielded the correct sequences without any mutations or deletions” – Would you not expect some mutations since you are using PCR to amplify fragments?

b. Why there are only two biological replicates for figure 2A? Legend is a missing description – in B – how many colonies did you check?, C – what was the template to amplify this mCherry gene?

c. The last concluding paragraph is too general. It is not true that by just having more plasmid DNA it will be possible to deliver DNA to “hard-to-transform” bacteria. Restriction-modification systems should be discussed more. What are the sizes of plasmids that you could create using this method? What mutation rate would you expect?

Protocol - please make sure that all information is included: for example, steps 1/2 - how much template DNA did you use? was it plasmid, genomic DNA. What enzyme? etc.

7. PLOS authors have the option to publish the peer review history of their article (what does this mean?). If published, this will include your full peer review and any attached files.

Reviewer #1: No

Reviewer #2: No

---

## [Author Response · Author response to Decision Letter 0]

18 Jan 2023

Editor's comments:

>Editor’s comment: We note that the grant information you provided in the ‘Funding Information’ and ‘Financial Disclosure’ sections do not match. When you resubmit, please ensure that you provide the correct grant numbers for the awards you received for your study in the ‘Funding Information’ section.

>Response: The ‘Financial Disclosure’ has been modified as follows to better match the ‘Funding Information’ section:

“This work was supported by a the Deutsche Forschungsgemeinschaft (DFG) Research grant [Project # 455063657 - https://gepris.dfg.de/gepris/projekt/455063657] for M.B.A., the DFG Collaborative Research Centre, SFB 1027 [Project # 200049484 - https://gepris.dfg.de/gepris/projekt/466932240] for S.S. and the Leibniz-Gemeinschaft's Leibniz Science Campus on Living Therapeutic Materials [LifeMat - https://www.lsclifemat.de/] for S.D. The funders had no role in study design, data collection and analysis, decision to publish, or preparation of the manuscript.”

>Editor’s comment: Thank you for stating the following financial disclosure:

“This work was supported by a Research Grant from the Deutsche Forschungsgemeinschaft (DFG) [Project # 455063657 - https://gepris.dfg.de/gepris/projekt/455063657] for M.B.A., the Collaborative Research Centre, SFB 1027 [Project # 200049484 - https://gepris.dfg.de/gepris/projekt/466932240] for S.S. and the Leibniz Science Campus on Living Therapeutic Materials [LifeMat - https://www.lsclifemat.de/] for S.D.”

>Response: This statement has been added in the ‘Financial Disclosure’ section as mentioned in the answer to the previous question.

>Editor’s comment: We note that you have stated that you will provide repository information for your data at acceptance. Should your manuscript be accepted for publication, we will hold it until you provide the relevant accession numbers or DOIs necessary to access your data. If you wish to make changes to your Data Availability statement, please describe these changes in your cover letter and we will update your Data Availability statement to reflect the information you provide.

>Response: All raw data including images and sequencing files related to results described in this manuscript have been added to the OSF data repository and will be made accessible once the manuscript is accepted for publication. Till then, a view only link is provided here - https://osf.io/c6x3d/?view_only=fbb9aede3b22455799b81f23a6270c65

>Editor’s comment: PLOS ONE now requires that authors provide the original uncropped and unadjusted images underlying all blot or gel results reported in a submission’s figures or Supporting Information files. This policy and the journal’s other requirements for blot/gel reporting and figure preparation are described in detail at https://journals.plos.org/plosone/s/figures#loc-blot-and-gel-reporting-requirements and https://journals.plos.org/plosone/s/figures#loc-preparing-figures-from-image-files. When you submit your revised manuscript, please ensure that your figures adhere fully to these guidelines and provide the original underlying images for all blot or gel data reported in your submission. See the following link for instructions on providing the original image data: https://journals.plos.org/plosone/s/figures#loc-original-images-for-blots-and-gels. In your cover letter, please note whether your blot/gel image data are in Supporting Information or posted at a public data repository, provide the repository URL if relevant, and provide specific details as to which raw blot/gel images, if any, are not available. Email us at plosone@plos.org if you have any questions.

>Response: All raw images of gels are provided in the data repository mentioned in the previous answer

>Editor’s comment: Please review your reference list to ensure that it is complete and correct. If you have cited papers that have been retracted, please include the rationale for doing so in the manuscript text, or remove these references and replace them with relevant current references. Any changes to the reference list should be mentioned in the rebuttal letter that accompanies your revised manuscript. If you need to cite a retracted article, indicate the article’s retracted status in the References list and also include a citation and full reference for the retraction notice.

>Response: The reference list has been checked and additional references after revision of the manuscript have been added and highlighted in the revised manuscript with track changes. To our knowledge we have not cited papers that were retracted. The following references have been added during the revision:

40. Potapov, V. and Ong, J.L., Examining sources of error in PCR by single-molecule sequencing. PloS one. 2017, 12(1), p.e0169774.

41. Karlskås, I.L., Maudal, K., Axelsson, L., Rud, I., Eijsink, V.G. and Mathiesen, G. Heterologous protein secretion in lactobacilli with modified pSIP vectors. PLOS one. 2014 9(3), p.e91125.

42. Johnston, C.D., Cotton, S.L., Rittling, S.R., Starr, J.R., Borisy, G.G., Dewhirst, F.E. and Lemon, K.P. Systematic evasion of the restriction-modification barrier in bacteria. Proceedings of the National Academy of Sciences, 2019 116(23), pp.11454-11459.

43. Mathiesen, G., Øverland, L., Kuczkowska, K. and Eijsink, V.G. Anchoring of heterologous proteins in multiple Lactobacillus species using anchors derived from Lactobacillus plantarum. Scientific Reports. 2020 10(1), pp.1-10.

Reviewer's comments:

Reviewer 1:

>Reviewer’s comment: The manuscript by Asensio et al, is really a fantastic example on the optimization of cloning methodologies to improve both cloning and transformation efficiencies with a wide range of heterologous expression and biochemical applications. The manuscript is very clear and well written. I recommend the manuscript for publication after some minor comments have been addressed.

>Response: We thank the reviewer for the positive remarks.

>Reviewer’s comment: The introduction is well written but some more introductory information on recently developed similar cloning strategies and how this method might provide further advantages (for example, increased DNA yield, ability to skip E. coli for toxic genes etc).

>Response: We thank the reviewer for this perspective. We tried to keep the introduction focused on the most important aspects directly related to the method we have described. We have now revised the introduction and included some more information and explanations. We expect the current introduction to have a better coverage of cloning protocols reported in literature as well as simultaneously highlighting the benefit of our work. 

>Reviewer’s comment: The authors do not do a DpnI digestion during the direct cloning approach and it is possible for the original template DNA to make it to the beginning of their transformation step (Day 2). Although the majority of this will be lost during the subsequent purification and PCR steps, did the authors have any problems with the background empty template being transformed? Did they find empty vectors in their colony screen? 

>Response: We acknowledge that prior to setting up the Gibson Assembly reaction in the direct cloning protocol, we did not conduct a DpnI digestion of the backbone vector template. This step is normally conducted for indirect cloning protocol mediated through E. coli DH5α in order to reduce the number of clones containing the circular vector backbone. 

We assumed that the undigested circular backbone vector would be significantly diluted during the multiple purification processes in the direct cloning protocol to produce negative clones in L. plantarum. Additionally, PCR screening of randomly selected transformants showed presence of the gene of insert confirming that clones do contain the recombinant plasmid and not the empty backbone vector.

However, following the reviewer’s suggestion, we repeated the experiment for constructing the pTlpA-mCherry plasmid by conducting a DpnI digestion of the vector backbone template before the Gibson Assembly reaction. After transformation, we verified by PCR that the red colonies (producing the desired phenotype) carry the assembled plasmid after the HiFi reaction and not the vector backbone. We screened 10 colonies for this experimental setup and 10 through the non-DpnI digested experimental setup. We got amplification at the expected size (558 bp) for both conditions. This helped us confirm our assumption that DpnI digestion of the backbone vector template does not provide significant advantage against the removal of empty clones after transformation. This data can be found in Figure S3.

Concerning this experiment, we have added the following text to the manuscript: 

“Furthermore, we wanted to prove that a DpnI digestion prior to the Gibson HiFi Assembly reaction would not have an impact on the number of positive clones obtained through the direct cloning method since we assumed that the backbone vectors used as template for the PCR would be extremely diluted during the multiple purification steps involved in the protocol. Therefore, we repeated the pLp_mCherry direct cloning with and without a DpnI digestion of the insert and vector PCR products prior to the Gibson HiFi Assembly reaction. We screened 10 red colonies for both experimental conditions by PCR amplification of a partial gene segment within the mCherry reporter. The proportion of red colonies obtained was similar in both conditions and colony PCR amplification of a part of the mCherry gene (10 red colonies from each) yielded PCR products of the same expected size from all clones. This confirmed that a DpnI digestion prior to the HiFi Assembly reaction is not necessary (Supporting Information Figure S3).”

>Reviewer’s comment: Could the authors comment on the proportion of positive vs negative clones during their colony screening and subsequent verification of plasmids in the final stages of their experiments?

>Response: Regarding positive and negative clones during colony screening, we found that all colonies that were red due to mCherry expression yielded positive clones on sequencing. This had been briefly mentioned in the manuscript with the following text:

“As expected, all clones expressing mCherry yielded the correct sequences without any mutations or deletions.”

On average, the number of red to non-red colonies that resulted from direct transformation was 83%. Figure 2B has now been updated in the manuscript. 

Based on the reviewer’s comment, we further screened 10 non-red colonies by amplifying the whole mCherry gene by colony PCR. The expected band should be 1140 bp. We got amplification (data can be found in Figure S4) for 7 out of 10 colonies. However, for most of them the amplicon was bigger than expected, suggesting the possibility that a minor proportion of wrongly assembled products were formed during Gibson assembly or random mutations were introduced during the PCR reaction. 

This is the text we have added to the manuscript addressing the screening of non-red clones. 

“The whole mCherry gene was also amplified from 10 non-red colonies using the same set of primers and PCR conditions as in Figure 2C. We obtained amplification for 7 out of 10, nevertheless, the PCR product was either bigger or smaller than expected (Supporting Information Figure S4). These results suggest that mutations might have occurred during the PCR amplification steps or a minor proportion of wrongly assembled products were formed during Gibson assembly which can result in recombinant clones with the mutated gene of interest (GOI).”

>Reviewer’s comment: The schematic figure (Figure 3) is great to show an overview of both methods. Could the authors also include slightly more information such as the strain they are performing their transformation into (cloning strain/expression strain etc). This should help to simplify the advantages of one method over another.

>Response: We thank the reviewer for their suggestion. We have made modifications in the Figure 3 scheme to incorporate details about the cloning strains during the bacterial transformation steps. We have also modified the scheme of Figure 1 in order to highlight the major attributes of the cloning procedure in further detail. 

>Reviewer’s comment: The authors are essentially performing PCRs on PCR products and thus the higher number of cycles eventually leads to the increased chance of incorporating mutations. We use Q5 a lot and understand that it is high fidelity and that mutations are very infrequent, also that the authors performed sequencing to ensure no errors. However, I think it would be nice if an extra sentence or two was placed in the text to ensure it is clear to both the authors and future scientists performing this method are aware of the issue and how it should be handled.

>Response: We thank the reviewer for their suggestion. We have added the following text in the newly written “Conclusions” section, highlighting the need for using a high-fidelity polymerase when following this protocol to minimize mutation risks as much as possible. 

“Since this method relies on PCR-amplification based in vitro assembly of DNA fragments, it must be noted that the accuracy can be affected by mutations that occur during PCR amplification and the possible formation of unspecific assembly fragments. To minimize the risk of mutations, a high-fidelity polymerase (Q5 DNA polymerase) was used in this study [40].”

>Reviewer’s comment: I think the conclusion is a bit weak and the authors could really use this opportunity to highlight the advantages of the method over what is currently available – particularly to the great biotechnology/synthetic biology fields.

>Response: We acknowledge the need for a separate “Conclusion” subheading in the manuscript. We have used this section to highlight the advantages of this cloning methodology for different applications in the field of synthetic biology. The revised text now includes the following statements:

“The direct cloning method developed in this paper has proved effective in transforming circular dsDNA (plasmid) DNA into L. plantarum WCFS1 without any intermediate host requirement for plasmid amplification. We demonstrate that this method provides two major benefits for lactobacillus engineering – (i) it saves time of at least 2 days compared to commonly used indirect cloning methods involving intermediate hosts and (ii) enables the cloning of genetic constructs that might be toxic or incompatible with the intermediate host. Since this method relies on PCR-amplification based in vitro assembly of DNA fragments, it must be noted that the accuracy can be affected by mutations that occur during PCR amplification and the possible formation of unspecific assembly fragments. To minimize the risk of mutations, a high-fidelity polymerase (Q5 DNA polymerase) was used in this study [40]. To accelerate the identification of positive colonies, a visible reporter like mCherry can be included or colony PCR can be performed. Using these methods, we confirmed that the accuracy of the transformed clones was above 80%. While we have tested this direct cloning method only in L. plantarum WCFS1, we believe this strategy can also be expanded to other hard-to-transform lactobacilli, in which similar plasmids have been previously transformed using the indirect method [41]. When testing the direct cloning method on different strains, it is important to note that success will depend on whether they accept unmethylated DNA. Furthermore, if transformation is hindered by restriction-modification systems in these strains, DNA design strategies can be employed to overcome this challenge [42]. Finally, while we have used modest-sized plasmid (<4 kb) with a low copy number replicon (P256 replicon, copy number 3 – 5). Based on previous studies [43], it is expected that bigger plasmids with higher copy number replicons can be transformed using the direct cloning method although further investigations are definitely needed to test the effect of plasmid size on efficiency of transformation and accuracy of the transformed clones.”

Reviewer 2:

>Reviewer’s comment: An article by Blanch-Asensio et al. “Gibson Assembly-based direct cloning of plasmid DNA in Lactiplantibacillus plantarum WCSF1” describes a modified method to deliver plasmids to L. Plantarum. This manuscript needs major revisions before it can be published.

>Response: We acknowledge the suggestions of the reviewer and have made substantial revisions to the manuscript. The changes have been highlighted in the manuscript for faster identification.

>Reviewer’s comment: The title of the paper gives off the impression that the authors successfully demonstrated in vivo DNA assembly. But the paper is just about obtaining higher plasmid copy numbers before the transformation.

>Response: We agree with the concern with the reviewer regarding the title of the manuscript. Our protocol mainly focusses on the in vitro assembly of circular plasmid DNA prior to the transformation in L. plantarum competent cells and does not rely on any in vivo assembly of gene segments in the intracellular milieu of the bacteria. To avoid misleading the readers in any form we have changed the title of our manuscript to “In vitro assembly of plasmid DNA for direct cloning in Lactiplantibacillus plantarum WCSF1”. 

>Reviewer’s comment: Introduction is lacking some information/explanations.

>Response: We thank the reviewer for this perspective. We tried to keep the introduction focused on the most important aspects directly related to the method we have described. We have now revised the introduction and included some more information and explanations. We expect the current introduction to have a better coverage of cloning protocols reported in literature as well as simultaneously highlighting the benefit of our work. 

Apart from the introduction, we have included a dedicated Conclusions section that summarizes our results in a broader context with additional information from literature.

>Reviewer’s comment: Some sentences are not clear “Hence, it is desirable to be able to clone these lactobacilli without the need for an intermediate host” - deliver plasmids? 

>Response: We agree with the reviewer that the sentence construction for the statement seems ambiguous. We have corrected the statement to convey the message more clearly:

“Hence, it is desirable to be able to directly transform circular plasmid dsDNA into the lactobacilli strains without relying on intermediate bacterial hosts like E. coli and L. lactis.”

>Reviewer’s comment: Why rolling circle amplification is not described? The is a great example of using this method for creating synthetic minimal cells -https://www.science.org/doi/10.1126/science.aad6253. How would this method compare with the method described in this manuscript?

>Response: We thank the reviewer for bringing up this complementary technique for DNA amplification. In this study, we focused on employing the most common methods used by the lactobacilli community so that they can easily adopt the method we developed. Rolling Circle Amplification is a powerful technique to greatly amplify DNA from very low template concentrations and has been used to solve complex challenges like those faced in the generation of minimal bacterial genomes. However, this method also faces several limitations like the need for specific sequences on the template DNA for the recognition of the nicking enzyme, non-specific amplification that can lead to false positives and the need for restriction enzymes to break up the long DNA fragment into individual plasmids. For these reasons, this method is not commonly used in bacterial engineering.

Accordingly, we also do not have practical experience with this method and any comparison we can make with our method would be purely speculative. Thus, we consider that the inclusion of a discussion based on RCA will be beyond the scope of this study and distract the reader from the main message of the manuscript.

>Reviewer’s comment: “As expected, all clones expressing mCherry yielded the correct sequences without any mutations or deletions” – Would you not expect some mutations since you are using PCR to amplify fragments?

>Response: We agree with the concerns of the reviewer in terms of spontaneous mutations being generated while amplifying the assembled gene fragments by PCR. In order to avoid the generation of mutations in these cases, we have chosen the high fidelity Q5 DNA Polymerase enzyme for minimizing the error rate for getting the correct amplicon. The most common form of error introduction is attributed to base substitution which has been reported to be the lowest for the Q5 polymerase (Potapov et al., 2017). 

Despite that, there obviously lies a chance of generating some mutations in the gene of interest which might be passed on to the recombinant clones after successful transformation. It should be noted that sequencing was done only for red clones which provided correct-sized bands post colony PCR analysis. In these cases, it was expected that no mutations would have occurred, which is the basis of the statement mentioned by the reviewer. However, it should be noted that about 80% of the colonies were red (Figure 2B), suggesting that mutations or assembly errors might have occurred in the remaining transformants. We have now performed colony PCR with those non-red colonies and confirmed that the amplified products are bigger or smaller than what is expected (Figure S4). We have further included text to clarify the possibility of mutations or assembly errors and in the Conclusions:

“These results suggest that mutations might have occurred during the PCR amplification steps or a minor proportion of wrongly assembled products were formed during Gibson assembly.”

Conclusions – “Since this method relies on PCR-amplification based in vitro assembly of DNA fragments, it must be noted that the accuracy can be affected by mutations that occur during PCR amplification and the possible formation of unspecific assembly fragments. To minimize the risk of mutations, a high-fidelity polymerase (Q5 DNA polymerase) was used in this study [40]. To accelerate the identification of positive colonies, a visible reporter like mCherry can be included or colony PCR can be performed. Using these methods, we confirmed that accuracy of the transformed clones was above 80%.”

References:

1. Potapov, V. and Ong, J.L., 2017. Examining sources of error in PCR by single-molecule sequencing. PloS one, 12(1), p.e0169774.

>Reviewer’s comment: Why there are only two biological replicates for figure 2A? Legend is a missing description – in B – how many colonies did you check?, C – what was the template to amplify this mCherry gene?

>Response: We acknowledge the comments of the reviewer. We have performed a third replicate and updated the graphs of Figure 2A and 2B. In the accuracy graph (Figure 2B), we checked 54 colonies for the 300 ng transformations (41 were red; 75.92% accuracy), 89 for the 600 ng transformations (83 were red; 93.25% accuracy), 133 for the 900 ng transformations (105 were red; 78.94 % accuracy), and 142 for the 1200 ng transformations (118 were red; 83.09% accuracy). Overall, 347 out of 418 colonies were red, which accounted for 83.01% accuracy. We have added these details to the manuscript:

“Colonies were streaked on fresh plates and the following day they were examined for the presence of fluorescent protein. Overall, 347 out of 418 colonies were red, giving an accuracy of 83% (Figure 2B).”

For figure 2C, we used a red colony as template for the colony PCR. We have added this detail to the legend:

“Agarose gel showing the colony PCR product (1140 bp) corresponding to the mCherry gene of interest (GOI). A red L. plantarum colony obtained after direct cloning was used as the template DNA for the PCR reaction. Generuler 100 bp Plus DNA Ladder (ThermoFisher ScientificTM) was used for the reference.”

Additionally, we have added an extra figure, Figure S4. We thought that it was interesting to amplify the whole mCherry gene from non-red colonies so we could get some insight into why these clones were not red. We screened 10 non-red colonies by performing the same PCR as in Figure 2C. Out of 10 colonies, we got clear amplification for 7. However, the PCR product was either bigger or shorter than expected (Figure 2C). These results possibly point out that mutations were introduced during the PCR amplification or wrongly assembled products were formed during Gibson assembly. 

Regarding this additional experiment, we have added the following text to the manuscript:

“The whole mCherry gene was also amplified from 10 non-red colonies using the same set of primers and PCR conditions as in Figure 2C. We obtained amplification for 7 out of 10, nevertheless, the PCR product was either bigger or smaller than expected (Supporting Information Figure S4). These results suggest that a minor proportion of wrongly assembled products were formed during Gibson assembly which can result in recombinant clones with the mutated gene of interest (GOI).”

>Reviewer’s comment: The last concluding paragraph is too general. It is not true that by just having more plasmid DNA it will be possible to deliver DNA to “hard-to-transform” bacteria. Restriction-modification systems should be discussed more. What are the sizes of plasmids that you could create using this method? What mutation rate would you expect? 

>Response: We acknowledge that the concluding paragraph was too generalized and did not provide sufficient coverage to related aspects of bacterial transformation. We have now used the “Conclusions” section in the manuscript to provide further insights in this regard. The revised text now includes the following statements:

“The direct cloning method developed in this paper has proved effective in transforming circular dsDNA (plasmid) DNA into L. plantarum WCFS1 without any intermediate host requirement for plasmid amplification. We demonstrate that this method provides two major benefits for lactobacillus engineering – (i) it saves time of at least 2 days compared to commonly used indirect cloning methods involving intermediate hosts and (ii) enables the cloning of genetic constructs that might be toxic or incompatible with the intermediate host. Since this method relies on PCR-amplification based in vitro assembly of DNA fragments, it must be noted that the accuracy can be affected by mutations that occur during PCR amplification and the possible formation of unspecific assembly fragments. To minimize the risk of mutations, a high-fidelity polymerase (Q5 DNA polymerase) was used in this study [40]. To accelerate the identification of positive colonies, a visible reporter like mCherry can be included or colony PCR can be performed. Using these methods, we confirmed that the accuracy of the transformed clones was above 80%. While we have tested this direct cloning method only in L. plantarum WCFS1, we believe this strategy can also be expanded to other hard-to-transform lactobacilli, in which similar plasmids have been previously transformed using the indirect method [41]. When testing the direct cloning method on different strains, it is important to note that success will depend on whether they accept unmethylated DNA. Furthermore, if transformation is hindered by restriction-modification systems in these strains, DNA design strategies can be employed to overcome this challenge [42]. Finally, while we have used modest-sized plasmid (<4 kb) with a low copy number replicon (P256 replicon, copy number 3 – 5). Based on previous studies [43], it is expected that bigger plasmids with higher copy number replicons can be transformed using the direct cloning method although further investigations are definitely needed to test the effect of plasmid size on efficiency of transformation and accuracy of the transformed clones.”

Reviewer’s comment: Protocol - please make sure that all information is included: for example, steps 1/2 - how much template DNA did you use? was it plasmid, genomic DNA. What enzyme? Etc.

Response: The protocol shared in the public repository has been written to provide a detailed overview of the cloning strategy. We had generalized some technical details in the protocol to provide flexibility to readers while designing individual experiments. Therefore, we had not specified the origin of the template DNA in the original version (e.g., plasmid, genomic DNA, synthetic gene). 

However, we acknowledge the reviewer’s suggestions to provide further details to ensure the protocol’s reproducibility. In step 1 (Molecular Cloning Part) we have highlighted the imperative use of a high-fidelity polymerase for the reaction. In steps 1 and 2 (Molecular Cloning Part) we have also specified the amount of template DNA we recommend using (10 ng). In step 2 (Molecular Cloning Part) we have added the specific requirements pertaining to the primer overhang lengths for ensuring ideal assembly. Finally, we have also included additional details in step 6 (Molecular Cloning Part) specifying the importance of an impeccable design of primers, suggesting a thorough analysis of the primers needed for the amplification of the assembled DNA with a primer design tool. 

This is the link to the updated Protocol:

https://www.protocols.io/private/CD173FD23D7711EDB7F00A58A9FEAC02

---

## [Editor Report · Decision Letter 1]

30 Jan 2023

In vitro assembly of plasmid DNA for direct cloning in Lactiplantibacillus plantarum WCSF1

PONE-D-22-25198R1

Dear Dr. Sankaran,

We’re pleased to inform you that your manuscript has been judged scientifically suitable for publication and will be formally accepted for publication once it meets all outstanding technical requirements.

Kind regards,

Hari S. Misra

Academic Editor

PLOS ONE
---

## [Editor Report · Acceptance letter]

7 Feb 2023

PONE-D-22-25198R1 

*In vitro* assembly of plasmid DNA for direct cloning in *Lactiplantibacillus plantarum* WCSF1 

Dear Dr. Sankaran:

I'm pleased to inform you that your manuscript has been deemed suitable for publication in PLOS ONE. Congratulations! Your manuscript is now with our production department. 

Kind regards, 

on behalf of

Professor Hari S. Misra 

Academic Editor

PLOS ONE